# Hemolysis Index Correlations with Plasma-Free Hemoglobin and Plasma Lactate Dehydrogenase in Critically Ill Patients under Extracorporeal Membrane Oxygenation or Mechanical Circulatory Support—A Single-Center Study

**DOI:** 10.3390/diagnostics14070680

**Published:** 2024-03-23

**Authors:** Bernhard Zapletal, Daniel Zimpfer, Thomas Schlöglhofer, Monika Fritzer-Szekeres, Thomas Szekeres, Martin H. Bernardi, Johannes Geilen, Marcus J. Schultz, Edda M. Tschernko

**Affiliations:** 1Department of Anaesthesiology, General Intensive Care and Pain Medicine, Division of Cardiac Thoracic Vascular Anesthesia and Intensive Care Medicine, Medical University Vienna, 1090 Vienna, Austria; bernhard.zapletal@meduniwien.ac.at (B.Z.); martin.bernardi@meduniwien.ac.at (M.H.B.); johannes.geilen@meduniwien.ac.at (J.G.); edda.tschernko@meduniwien.ac.at (E.M.T.); 2Department of Cardiac Surgery, Medical University Vienna, 1090 Vienna, Austria; daniel.zimpfer@meduniwien.ac.at (D.Z.); thomas.schloeglhofer@meduniwien.ac.at (T.S.); 3Center for Medical Physics and Biomedical Engineering, Medical University Vienna, 1090 Vienna, Austria; 4Department of Laboratory Medicine, Medical University Vienna, 1090 Vienna, Austria; monika.fritzer-szekeres@meduniwien.ac.at (M.F.-S.); thomas.szekeres@meduniwien.ac.at (T.S.); 5Department of Intensive Care, Amsterdam University Medical Center, 1105 AZ Amsterdam, The Netherlands

**Keywords:** intensive care, mechanical circulatory support, extracorporeal life support, ELS, extracorporeal membrane oxygenation, ECMO, mechanical circulatory support, hemolysis index, HI, plasma-free hemoglobin, PFH, lactate dehydrogenase, LDH

## Abstract

Monitoring for thrombosis and hemolysis is crucial for patients under extracorporeal or mechanical circulatory support, but it can be costly. We investigated correlations between hemolysis index (HI) and plasma-free hemoglobin (PFH) levels on one hand, and between the HI and plasma lactate dehydrogenase (LDH) levels on the other, in critically ill patients with and without extracorporeal or mechanical circulatory support. Additionally, we calculated the cost reductions if monitoring through HI were to replace monitoring through PFH or plasma LDH. In a single-center study, HI was compared with PFH and plasma LDH levels in blood samples taken for routine purposes in critically ill patients with and without extracorporeal or mechanical circulatory support. A cost analysis, restricted to direct costs associated with each measurement, was made for an average 10-bed ICU. This study included 147 patients: 56 patients with extracorporeal or mechanical circulatory support (450 measurements) and 91 patients without extracorporeal or mechanical circulatory support (562 measurements). The HI correlated well with PFH levels (r = 0.96; *p* < 0.01) and poorly with plasma LDH levels (r = 0.07; *p* < 0.01) in patients with extracorporeal or mechanical circulatory support. Similarly, HI correlated well with PFH levels (r = 0.97; *p* < 0.01) and poorly with plasma LDH levels (r = −0.04; *p* = 0.39) in patients without extracorporeal or mechanical circulatory support. ROC analyses demonstrated a strong performance of HI, with the curve indicating excellent discrimination in the whole cohort (area under the ROC of 0.969) as well as in patients under ECMO or mechanical circulatory support (area under the ROC of 0.988). Although the negative predictive value of HI for predicting PFH levels > 10 mg/dL was high, its positive predictive value was found to be poor at various cutoffs. A simple cost analysis showed substantial cost reduction if HI were to replace PFH or plasma LDH for hemolysis monitoring. In conclusion, in this cohort of critically ill patients with and without extracorporeal or mechanical circulatory support, HI correlated well with PFH levels, but poorly with plasma LDH levels. Given the high correlation and substantial cost reductions, a strategy utilizing HI may be preferable for monitoring for hemolysis compared to monitoring strategies based on PFH or plasma LDH. The PPV of HI, however, is unacceptably low to be used as a diagnostic test.

## 1. Introduction

Extracorporeal membrane oxygenation (ECMO) and mechanical circulatory support using right or left ventricular assist devices (VADs) are life-saving strategies in patients suffering from end-stage heart failure. Pump thrombosis and thromboembolism are common complications [1,2,3,4] and systemic anticoagulation is imperative. The management of anticoagulation in patients under mechanical circulatory support can pose unique challenges, especially in the postoperative phase.

The timely assessment of thrombosis and hemolysis in patients under ECMO or mechanical circulatory support with VAD relies on the monitoring of specific laboratory parameters, such as plasma-free hemoglobin (PFH) and plasma lactate dehydrogenase (LDH). PFH levels serve as a valuable tool for the early detection of subclinical thrombosis [5] and pump thrombosis in patients under ECMO or mechanical circulatory support with VAD [5,6,7,8]. Additionally, plasma LDH levels are used to aid in the identification of hemolysis [9,10,11]. It is important to note that the measurements of both laboratory parameters come at a considerable cost, and the reliability of plasma LDH levels may be compromised in patients with hyperbilirubinemia, a condition often observed in patients undergoing mechanical circulatory support.

The hemolysis index (HI), a quantitative and qualitative parameter routinely assessed during laboratory blood analysis to evaluate the preanalytical state of blood samples [12], emerges as a promising alternative. The HI is a calculation based on absorbance measurements performed on serum/plasma at different wavelengths, providing a semi-quantitative estimate of hemolysis detected in a sample [13]. The HI not only serves laboratory purposes but also holds the potential to effectively replace measurements of PFH and plasma LDH in the clinic. Previous research indicates a strong correlation between the HI and PFH in both healthy volunteers as well as in patients under ECMO [12,14,15,16]. In addition, HI is a cost-free parameter and has the potential to result in substantial cost reductions when replacing PFH or plasma LDH for routine monitoring for hemolysis.

We conducted simultaneous pairwise measurements of the HI, PFH, and plasma LDH in blood samples obtained from critically ill patients, both with and without extracorporeal or mechanical circulatory support. Additionally, we performed a simple cost analysis. We hypothesized that HI correlates well with PFH levels and plasma LDH levels, justifying routine clinical use.

## 2. Methods

### 2.1. Design

This was a single-center observational cohort study in the cardiothoracic intensive care unit (ICU) of a university hospital. This study was performed in accordance with the Helsinki Declaration and approved by the Institutional Review Board (EC number 1213/2019).

The need for individual patient informed consent was waived as we used measurements that were taken for routine purposes.

### 2.2. Patients

Patients were eligible if (1) admitted to the ICU; (2) admitted for heart failure with or without extracorporeal life or mechanical circulatory support; and if (3) the HI, PFH, and plasma LDH were measured simultaneously.

Samples in which one of these parameters was not measured were excluded. The one single reason why HI and PFH were not measured in routinely obtained samples was a plasma bilirubin level > 60 mg/dL.

### 2.3. Data Collected

We collected demographic data and baseline characteristics of patients, including gender, age, reason for ICU admission, and disease severity score on admission. It was documented whether a patient was on ECMO or was receiving mechanical circulatory support from a right or left VAD. We also determined whether thromboembolic complications including pump thrombosis and stroke were present at the time of blood sampling.

### 2.4. Measurements

Blood, sampled for routine purposes, was drawn in siliconized vacuum tubes (Vacuette 9NC Coagulation sodium citrate 3.2%, Greiner Bio-One, Kremsmünster, Austria) and sent to the central laboratory. In patients without extracorporeal or mechanical circulatory support, HI, PFH levels, and plasma LDH levels were measured once per day; in patients under extracorporeal or mechanical circulatory support, more measurements per day were performed during the initial postoperative period.

HI was measured for laboratory quality control purposes by assessing the absorbance of light at 570 and 600 nm using the Roche Cobas C 702 (Roche, Basel, Switzerland). PFH levels were measured using the pseudoperoxidase method [17] and plasma LDH levels were measured by determining the catalytic activity via the reduction of NAD to NADH and photometric measurement (Roche Cobas C 702).

### 2.5. Endpoints

The primary endpoint was the correlation between the HI and the PFH levels. The secondary endpoint was the correlation between the HI and the plasma LDH levels.

### 2.6. Power Calculation

The HI, PFH, and plasma LDH were simultaneously measured for a period of 4 months. We expected that this would provide us with measurements in approximately 150 patients.

### 2.7. Statistical Analysis

Data are presented as means with standard deviation or medians with interquartile ranges where appropriate; categorical variables are presented in absolute values and relative proportions.

Correlations between the HI, PFH, and plasma LDH were calculated using Pearson correlation coefficients. A correlation coefficient above 0.9 and *p* < 0.05 was regarded as an excellent correlation of laboratory parameters [18]. Correlations were calculated and reported for patients with and without extracorporeal or mechanical circulatory support. We also calculated the correlation between the HI, PFH, and plasma LDH in patients with a high HI versus patients with a low HI, using a cutoff of 20 [14].

We performed Bland and Altman analyses for HI measurements ≤20 and >20, based on previous recommendations [19], and constructed receiver operating characteristic (ROC) curves to calculate sensitivity, specificity, and the positive and negative predictive value (PPV and NPV) of HI to detect hemolysis defined by PFH > 10 mg/dl [19].

Next, we compared the laboratory costs using in-hospital billing for the three laboratory parameters. Per measurement, the cost was EUR 40 for a single PFH measurement and EUR 4 for a single plasma LDH measurement, while HI was available free-of-charge as a routine preanalytical parameter. We extrapolated costs to a 10-bed ICU, wherein 8 beds were used for patients without mechanical circulatory support, i.e., patients that received hemolysis screening once every day, and 2 beds were for patients with extracorporeal or mechanical circulatory support, i.e., patients that received hemolysis screening four times every day. We calculated costs assuming an occupancy rate of 100%.

Data were collected and stored in a database created using Microsoft Excel. All statistical analyses were performed using Python 3.9.2 with Pandas 1.4.1 and SciPy 1.8.0 package (Python Software Foundation, Fredericksburg, VI, USA). Statistical significance was set at *p* < 0.05.

## 3. Results

### 3.1. Patients

From the 1 September to the 31 December 2018, 147 patients were included in this study: 3 patients under ECMO, 53 patients with LVAD or temporary RVAD support, and 91 patients without any of these supports (Figure 1).

Patients were predominately male (72%) and most patients were over 65 years old (Table 1). The primary reasons for extracorporeal or mechanical circulatory support were end-stage heart failure due to various reasons (idiopathic, myocarditis, coronary artery disease). The HI, PFH, and plasma LDH were simultaneously measured in a total of 1012 blood samples; of these, 120 (median 4 [3 to 4] per day) blood samples were drawn from patients under ECMO, 330 (median 3 [1 to 4] per day) were drawn from patients with a VAD, and 562 (median 1 [1 to 1] per day) were drawn from patients without extracorporeal or mechanical circulatory support.

### 3.2. Correlations between the HI and PFH Levels and Plasma LDH Levels

In the overall cohort, the correlation between the HI and PFH levels was excellent (r = 0.97; *p* < 0.01) (Figure 2 and Table 2) and not different for patients with (r = 0.96; *p* < 0.01) and without extracorporeal or mechanical circulatory support (r = 0.97; *p* < 0.01). In contrast, the correlation between the HI and plasma LDH levels was poor (r = 0.03; *p* = 0.33) in the overall cohort, and also in patients with (r = 0.10; *p* < 0.01) and without extracorporeal or mechanical circulatory support (r = −0.04; *p* = 0.39). A noticeable poor correlation was also observed between the PFH and plasma LDH levels. The findings were not different for patients with a high HI and patients with a low HI, using a cutoff of 20.

### 3.3. Bland and Altman Analyses

Bland and Altman analysis plots are shown in Figure 3. For samples with an HI ≤ 20, bias was 0.3 ± 0.2 and the upper and lower limits of agreement were 5.2 ± 0.2 and 4.6 ± 0.3. For samples with an HI > 20, bias was −16.04 ± 3.4 and the upper and lower limits of agreement were 13.8 ± 5.9 and −45.9 ± 5.9.

### 3.4. ROC Curves

ROC curves are shown in Figure 4. The ROC analyses demonstrated a strong performance, with the curve indicating excellent discrimination, in the whole cohort as well as in patients under ECMO or mechanical circulatory support. Test characteristics are presented in Table 3. Using a cutoff for HI of 8.5 resulted in a sensitivity and a specificity of 100% and 88.8% and an excellent NPV but a poor PPV of 100% and 37.8%, respectively, in patients with ECMO or mechanical circulatory support. Increasing the cutoff to 9.5 improved the specificity and PPV somewhat, but at the price of lower sensitivity and a lower NPV in these patients.

### 3.5. Cost Calculation

In our cohort, the daily costs for measurements of HI, PFH, and LDH were median EUR 0 [0 to 0], EUR 40 [40 to 40], and EUR 4 [4 to 4] for patients without extracorporeal or mechanical circulatory support, and median EUR 0 [0 to 0], EUR 160 [120 to 160], and EUR 16 [12 to 16] for patients with extracorporeal or mechanical circulatory support (Table 4). Extrapolating costs to a hypothetical 10-bed ICU, adopting a strategy using HI instead of FPH or plasma LDH, would yield a monthly cost reduction of a median EUR 19,200 [EUR 16,800 to EUR 19,200] or a median EUR 1920 [EUR 1680 to EUR 1920].

## 4. Discussion

The findings of this single-center observational study can be summarized as follows: (i.) HI and PFH levels have an excellent correlation, both in patients with extracorporeal or mechanical circulatory support and in patients not receiving one of these forms of support and (ii.) the correlation between the HI and plasma LDH levels is poor, in both patient groups. In addition, (iii.) the findings were not different for patients with a high HI and patients with a low HI. The Bland and Altman analysis provided a quantitative estimate of how closely the values of HI and PFH align. In samples with an HI ≤ 20, the findings suggest that HI and PFH can be used interchangeably. Sensitivity and NPV were excellent, in the entire cohort, as well as in the group with the highest risk of hemolysis. Specificity and PPV, however, were low, even after increasing the cutoff for HI. A simple cost calculation suggests a substantial cost-saving effect when replacing hemolysis monitoring through PFH or plasma LDH levels with monitoring through HI.

Our study has strengths. We included patients under various forms of extracorporeal and mechanical circulatory support, and patients receiving neither extracorporeal nor mechanical circulatory support, furthering the generalizability of our findings. This was a single-center study, but our center is an expertise center for the care of patients receiving extracorporeal or mechanical circulatory support. We used blood samples that were taken for routine measurement of PFH and plasma LDH levels, reflecting our daily practice. All measurements were performed in a standard way in the central laboratory of our hospital. Measurements were performed batchwise, meaning that the HI, PFH, and plasma LDH levels always related in time. Last but not least, we used real costs, i.e., costs charged by the central laboratory, for cost comparisons.

The original purpose of HI is to detect hemolysis which may interfere with certain analyses, i.e., it is meant for use in the laboratory to detect preanalytical impairments of samples. It is important to note that the degree of hemolysis susceptible to interference with downstream laboratory analysis is not the same as the degree of hemolysis susceptible to cause damages to patients. However, this laboratory parameter could be useful to alert clinicians that hemolysis is possibly occurring, which should trigger additional actions.

Our finding that HI correlates well with PFH levels confirms the findings of previous investigations [12,13,14,15]. A good correlation between the HI and PFH levels was found in plasma from healthy volunteers and in various laboratories using different blood chemistry analyzers [12]. The good correlation between the HI and PFH levels was confirmed in a small study using artificial plasma samples [16]. In clinical practice, the correlation between the HI and PFH levels was found to be good in pediatric ECMO patients [15]. Of note, in ECMO patients, a hemolysis index (HI) exceeding 20 is associated with increased mortality [14]. Our study extends previous findings, demonstrating excellent correlations between the HI and PFH levels in a large patient cohort with and without extracorporeal or mechanical circulatory support, irrespective of high or low HI levels.

Our finding that HI correlates poorly with plasma LDH levels contrasts findings of earlier studies [3,20,21]. A previous study showed an excellent correlation in intraoperative cell salvage solutions [21]. A good correlation was also found in a study comparing HI and plasma LDH levels with respect to predicting sickle cell crises [20]. LDH is a nonspecific enzyme, present in various cell types, and its levels rise during inflammation, ischemia, and organ damage. Plasma LDH levels can therefore be used for monitoring cell damage in general, rather than being specifically used for hemolysis [9,10,11]. Our findings challenge the definitions of hemolysis in the INTERMACS registry, wherein hemolysis is defined as PFH levels > 20 mg/dl or plasma LDH levels > 2.5 times the upper limit of the normal range [22].

A notable finding was the weak correlation between the PFH and plasma LDH levels. This finding is significant in light of previous research associating elevated plasma LDH levels with pump thrombosis in LVAD patients [3,22]. However, we were unable to assess the accuracy of HI, PFH, and plasma LDH levels in routine monitoring for predicting or detecting ECMO or VAD pump thrombosis due to their rarity; indeed, these events did not occur in our cohort. Further investigation is needed.

The cost analysis revealed a significant economic advantage of using HI for monitoring compared to PFH and plasma LDH measurements. Given the high costs associated with patients under extracorporeal or mechanical circulatory support and increasing economic pressure [21], our analysis demonstrated a monthly cost reduction from around EUR 20,000 to zero for a fictive 10-bed unit by switching from PFH to HI. This strong shift was facilitated by the HI being provided free-of-charge by all machines conducting standard blood chemistry analysis. Subsequently, in collaboration with the department of laboratory medicine, we decided to substitute PFH with the HI for hemolysis monitoring.

In our ICU, we deem monitoring hemolysis an essential strategy for patients undergoing ECMO or mechanical circulatory support. We started using HI to evaluate the hemolysis status in these patients, as this measure came at no additional costs. Due to this convenience, this policy has been extended to patients weaned off ECMO or mechanical circulatory support, albeit with a lower frequency of measurements. We believe this practice is not unique to our center; however, it is evident that the cost-effectiveness may vary between centers when employing different sampling frequencies.

Based on our findings, PFH may be disregarded in the future, but LDH monitoring may have other clinical implications. Indeed, LDH may increase for various medical reasons, including liver disease, anemia, cardiac injury, muscle trauma, cancers, and infections. It is crucial to emphasize that while HI may serve as a monitoring tool for hemolysis, our findings unequivocally demonstrate its insufficiency as a diagnostic instrument. The inadequacy of HI as a reliable diagnostic tool is evident from our analysis, rendering it unreliable with a notably low PPV. Consequently, clinicians must exercise caution and refrain from solely relying on HI measurements in making critical clinical decisions.

Our study has limitations. In our study, we employed a single blood chemistry analyzer, the Roche Cobas C 702. It should be noted that HI measurements are not standardized across various laboratory platforms, potentially contributing to discrepancies in results across studies. While one study suggests that the variance in HI measurements could be acceptable [12], another study suggests the contrary [23]. In addition, the quality control of HI and calibration of analyzers can be limited by problems with commercially available quality control samples, another challenge that deserves a solution from the suppliers [16]. The reliability of HI measurements is impaired in cases of icteric and lipemic samples, conditions that are automatically displayed by the blood chemistry analyzer. In our analysis, we excluded patients with icterus, and none of the samples were lipemic. Pump thrombosis and other thromboembolic events did not occur, making clinical correlation of HI, PFH, and plasma LDH levels with such events impossible. In our study, we only used one specific blood chemistry analyzer. Future studies are needed to confirm our findings using other blood chemistry analyzers. Last but not least, this was a single-center study, in a unit with extensive experience in care for patients with ECMO and patients under mechanical circulatory support. Future studies are needed to confirm that HI is a good predictor for pump thrombosis and to confirm the overall cost-saving effects of replacing monitoring through PFH or plasma LDH levels with monitoring through HI.

## 5. Conclusions

In this cohort of critically ill patients, HI had an excellent correlation with PFH levels and a poor correlation with plasma LDH levels, both in patients with and without extracorporeal or mechanical circulatory support. The HI may be an attractive alternative for PFH or plasma LDH in respect of monitoring for hemolysis in ICUs. The PPV of HI, however, is unacceptably low to be used as a diagnostic test. A simple cost analysis showed substantial cost reductions if HI replaces PFH or plasma LDH for routine monitoring.

## Figures and Tables

**Figure 1 diagnostics-14-00680-f001:**
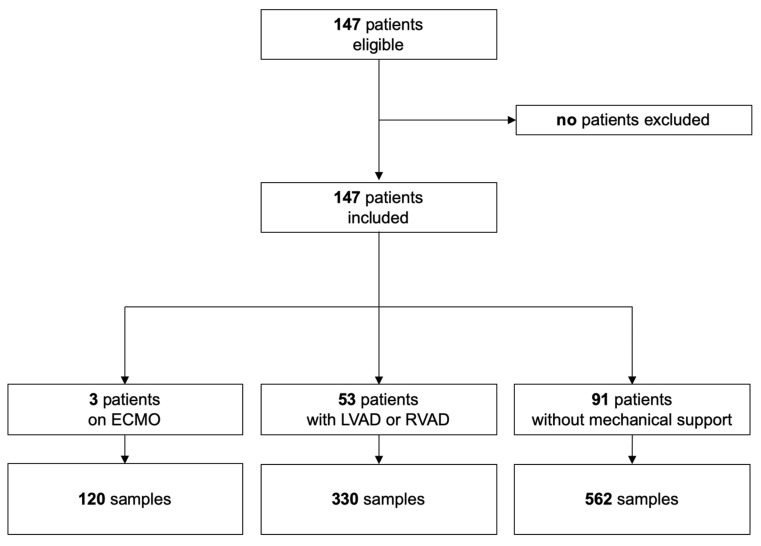
Flowchart of patients and numbers of samples. Abbreviations: ECMO, extracorporeal membrane oxygenation; LVAD, left ventricular assist device; RVAD right ventricular assist device.

**Figure 2 diagnostics-14-00680-f002:**
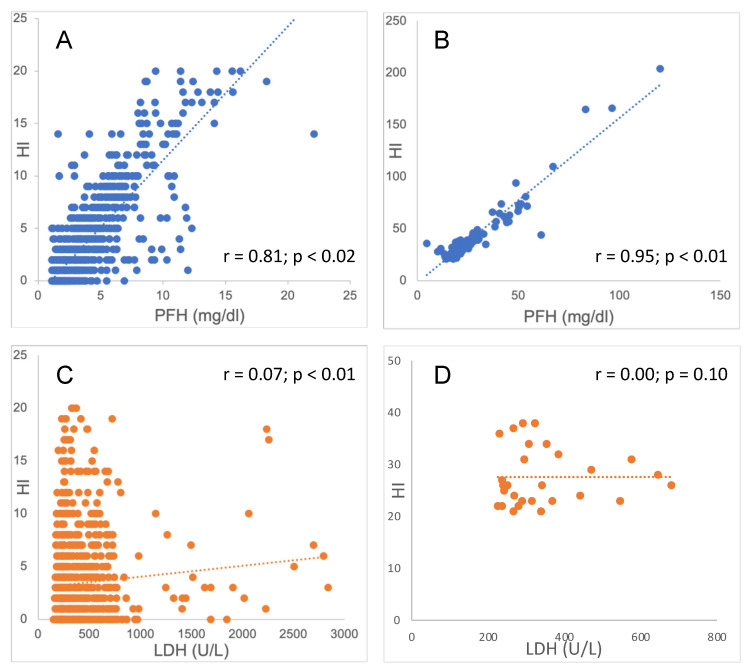
Correlations between the HI, PFH, and plasma LDH; dashed line: trendline; (panel **A**) correlation between the HI and PFH in samples with an HI below 20; (panel **B**) correlation diagram of HI and PFH in samples with an HI exceeding 20; (panel **C**) correlation diagram of HI and LDH in samples with an HI below 20; and (panel **D**) correlation diagram of HI and LDH in samples with an HI exceeding 20. Abbreviations: HI, hemolysis index; PFH, plasma-free hemoglobin; LDH, lactate dehydrogenase.

**Figure 3 diagnostics-14-00680-f003:**
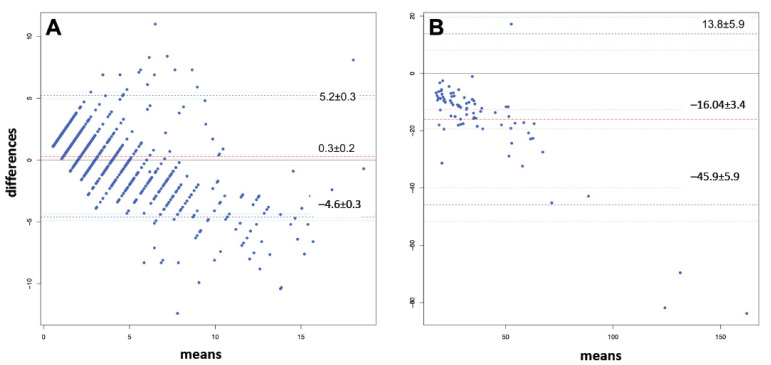
Bland–Altman analysis: (panel **A**) for samples with HI measurements ≤20; (panel **B**) for samples with HI measurements and >20.

**Figure 4 diagnostics-14-00680-f004:**
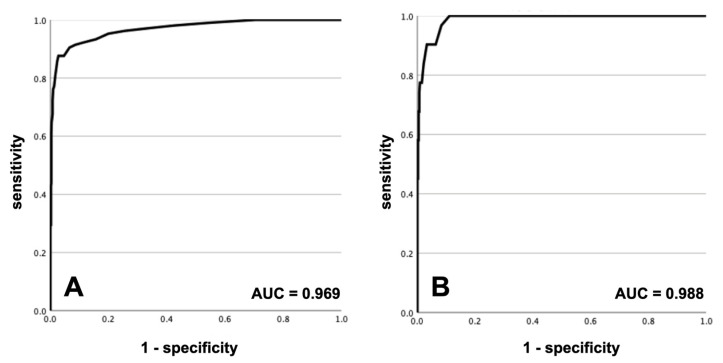
ROC curves: (panel **A**) for the whole cohort; (panel **B**) for patients with ECMO or mechanical circulatory support.

**Table 1 diagnostics-14-00680-t001:** Demographics and Baseline Characteristics.

	Patients with Extracorporeal or Mechanical Circulatory Support (*N* = 56)	Patients without Extracorporeal or Mechanical Circulatory Support (*N* = 91)
gender		
Female, n (%)	8 (14.3)	31 (34.1)
age, years, median [IQR]	58.3 [52.0–66.2]	74.9 [62.7–82.6]
body mass index, kg/m^2^, median [IQR]	28.1 [26.1–30.8]	27.0 [24.0–30.1]
reason for extracorporeal or mechanical circulatory support, n (%)		
coronary heart disease	–	20 (22.0)
aortic valve disease	–	27 (29.7)
mitral valve disease	–	6 (6.6)
tricuspid valve disease	–	3 (3.3)
multiple valve disease	2 (3.6)	12 (13.2)
cardiomyopathy	53 (94.6)	5 (5.5)
endocarditis	–	3 (3.3)
aortic aneurysm	–	4 (4.4)
aortic dissection	–	3 (3.3)
ventricular rupture	1 (1.8)	8 (8.8)
type of support (%)		
extracorporeal membrane oxygenation	3 (5.4)	–
mechanical circulatory support	53 (94.6)	–

Data are presented as numbers (%) or medians [IQR].

**Table 2 diagnostics-14-00680-t002:** Correlations between the measures.

	All Patients	Patients with Extracorporeal or Mechanical Circulatory Support	Patients without Extracorporeal or Mechanical Circulatory Support	Patients with HI ≤ 20	Patients with HI > 20
	(*N* = 147)	(*N* = 56)	(*N* = 91)	(*N* = 146)	(*N* = 64)
r, (Pearson) PFH/HI	0.97	0.96	0.97	0.81	0.95
P, PFH/HI	<0.01	<0.01	<0.01	0.02	<0.01
r, (Pearson) HI/plasma LDH	0.03	0.1	−0.04	0.07	0.00
P, HI/plasma LDH	0.33	<0.01	0.39	<0.01	0.1
r, (Pearson) PFH/plasma LDH	0.19	0.14	0.24	0.29	−0.17
P, PFH/plasma LDH	<0.01	<0.01	<0.01	<0.01	0.38

Abbreviations: HI = hemolysis index; PFH = plasma–free hemoglobin; LDH = lactate dehydrogenase.

**Table 3 diagnostics-14-00680-t003:** Test Characteristics of HI to predict PFH > 10, using 2 different cutoffs.

	Sensitivity	Specificity	PPV	NPV
Comparison hemolysis				
HI > 8.5 – PFH >10 (all)	100%	95.5%	76.8%	100%
HI > 8.5 – PFH >10 (VAD + ECMO)	100%	88.8%	37.8%	100%
HI > 9.5 – PFH >10 (all)	96.8%	91.7%	44.1%	99.8%
HI > 9.5 – PFH >10 (VAD + ECMO)	91.4%	91.4%	54.2%	99.0%

Data shown as % if stated; sensitivity and specificity of PFH is assumed as 100%; hemolysis measured by PFH if >10; no hemolysis if PFH ≤ 10; Abbreviations: PPV = positive predictive value; NPV = negative predictive value; HI = hemolysis index; PFH = plasma free hemoglobin.

**Table 4 diagnostics-14-00680-t004:** Cost analysis *.

	HI	PFH	Plasma LDH
Costs per measurement	€0	€40	€4
Costs per day			
patients with support	€0 [0–0]	€160 [120–160]	€16 [12–16]
ICU patients without support	€0 [0–0]	€40 [40–40]	€4 [4–40]
Costs per month for 10–bed ICU *	€0 [0–0]	€19,200 [16,800–19,200]	€1920 [1680–1920]

*, see text for details; values are medians [IQR]; Abbreviations: HI = hemolysis index; PFH = plasma free hemoglobin; LDH = lactate dehydrogenase; VAD = ventricular assist device; ICU = intensive care unit.

## Data Availability

The dataset of the current study is available upon request via the corresponding author.

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
