# Peer review of "Hemolysis Index Correlations with Plasma-Free Hemoglobin and Plasma Lactate Dehydrogenase in Critically Ill Patients under Extracorporeal Membrane Oxygenation or Mechanical Circulatory Support—A Single-Center Study"

_diagnostics, 2024, doi:10.3390/diagnostics14070680_

Round 1

Reviewer 1 Report

Comments and Suggestions for Authors

The topic and questions addressed by this work are important and deserve such investigations.

Nonetheless, it appears that major issues are present and preclude any robust conclusions, especially regarding the methodology used to formulate authors conclusions.

The main message of the paper is that quantification of hemolysis is correctly performed by the H-index (direct spectrophotometry-based measurements on samples), for free (automatic pre-analysis check by the automate) and could usefully replace costly method that determines plasma-free hemoglobin. In other words, the “H-index assay” provides a result “correlated” to the one obtained with pFH assay and should replace it (for cost reason namely).

I respectfully disagree with such a conclusion and I present my arguments below. Authors should rebuild their analysis using validated methods to support their message.

1-      Pearson correlation is not the correct methodology to assess the agreement between two analytical methods.

Even if the Pearson correlation coefficient is an easy and fast way to establish a mathematical relation between two methods, it is widely admitted that it is not a reliable tool to assess the equivalence (the agreement) between two methods [1-3].

 A perfect correlation could exist between two methods while they are totally not interchangeable.

I agree with the authors that HI and pFH correlates but this does not indicate that HI could replace pFH assay. Correlation is not agreement!

Others statistical methods like those described by Bland & Altman [4] or Lin [5] have to be used. In addition, diagnostic tests have to be characterized in terms of internal validity (sensitivity, specificity, positive predictive value, negative predictive value) and this part is totally lacking in the manuscript.

2-      H-index original purpose is not to precisely determine the concentration of plasma-free hemoglobin but rather to detect (with a good sensitivity) a level of hemolysis that is susceptible to negatively interfere with analysis performed downstream. The degree of hemolysis susceptible to interfere with downstream analysis in the lab is not the same that the degree of hemolysis susceptible to cause damages on patients.

This is why “H-index” is not called “assay for plasma free hemoglobin.” This is also why “H-index” has no In Vitro Diagnostic (IVD)-FDA or Conformity for Europe (CE) mark approval for determination of plasma-free hemoglobin.

3-      H-index, regarding its potential to quantify hemolysis in the ECMO/circulatory support settings has been associated with multiples drawbacks, preventing a simple and direct applicability to the clinical context. Sadely, your paper does not cite any reference dealing with such limits, leading to potentially biased overview and analysis of the question.

As example (see [6]):

- H-indexes are not comparable within manufacturers as remarkably demonstrated by Calvaresi et al. on Abbott, Roche, and Beckman platforms [7];

- H-index reliability is impaired in case of icteric and lipemic samples [8], both situations that are not exceptional in patients hospitalized in critical care unit with ECMO support (liver dysfunction, propofol infusion, parenteral nutrition);

- Some authors reported that H-index measurement demonstrated poor recovery when applied to commercial hemoglobin quality control material, questioning the ability of the method to be calibrated [9];

- H-index may have a very poor negative predictive value (NPV): with a cutoff set to 500 mg/L, Calvaresi et al. observed a NPV of 57.6% for one analyzer and 80.9% for another one [7]. Even if one could argue that this low NPV is here in relation with a high prevalence of the targeted anomaly (hemolysis), such performances are not suitable for a clinical use.

4-      When considering low level of hemolysis (H-index <20), data presented here clearly indicate a low agreement with pFH assay. For some patients (table A), a H-index value equal or lower to 15 could be associated with a pFH level close to zero or higher than 200 mg/L, definitely confirming that a robust and dedicated assay is better than a non-specific method, originally devoted to another purpose (i.e. to detect with a good sensitivity a level of hemolysis that is susceptible to interfere with analysis performed downstream);

5-      The design of the study is unusual to allow a robust analysis. Why did you include patients that are not at risk of mechanical hemolysis (no extra-corporeal or mechanical circulatory support)? This population represents here in your cohort a vast majority (62%) of the patients. In this population, by definition, hemolysis events are present at very low frequency. You then enter this population within the whole analysis, to assess the correlation of HI, pFH and LDH. By this way, you introduce a bias where you analyze couples of measures obtained from 2 different populations: one (minority) with a high prevalence of hemolysis and one (majority) with a low prevalence of hemolysis. As negative and positive predictive values depend on the prevalence of the disease in the studied population [10], such a design is ambiguous.

6-      In your paper, the performance of the H-index to correctly correlate with pFH is lower when <20 (+/- 200 mg/L in Roche automate), meaning that some patients (as previously exposed) could have no hemolysis or a significant hemolysis (>200 mg/L of pFH). ELSO guidelines [11] recommend to maintain pFH below 100 mg/L and Bosma et al. [12] observed that a repeated H-index >20 was a predictor of mortality. These 2 points clearly demonstrate that a H-index cut-off of 20 is too high (i.e. to belated) and that a more sensitive way of assessing hemolysis is required to prevent / treat (earlier) the causes of this hemolysis.

Minor remarks, corrections, questions or suggestions:

- Line 112. “using Pearson” should be replaced by “using Pearson coefficient/method”

- Line 119-124: I don’t understand the extrapolation that is performed. Why do you screen patients for hemolysis, once a day, when they are not receiving mechanical circulatory support? This is quite odd. In addition, why did you perform a monitoring of hemolysis every 6 hours (4 times per day) on patients at risk of hemolysis? Such a sampling frequency is quite high and unusual as a routine practice in our country and others. Could you justify such a habit? The cost reduction mentioned in your work is clearly impressive but it could be argued that you clearly did not apply the most economical strategy in routine.

References:

1.           Altman, D.G. and J.M. Bland, Assessing Agreement between Methods of Measurement. Clin Chem, 2017. 63(10): p. 1653-1654.

2.           van Stralen, K.J., et al., Agreement between methods. Kidney Int, 2008. 74(9): p. 1116-20.

3.           Watson, P.F. and A. Petrie, Method agreement analysis: a review of correct methodology. Theriogenology, 2010. 73(9): p. 1167-79.

4.           Bland, J.M. and D.G. Altman, Statistical methods for assessing agreement between two methods of clinical measurement. Lancet, 1986. 1(8476): p. 307-10.

5.           Lin, L.I., A concordance correlation coefficient to evaluate reproducibility. Biometrics, 1989. 45(1): p. 255-68.

6.           Dufour, N., H-index and Hemolysis Associated with ECMO: Is This So Simple? ASAIO J, 2021.

7.           Calvaresi, E.C., et al., Plasma hemoglobin: A method comparison of six assays for hemoglobin and hemolysis index measurement. Int J Lab Hematol, 2021.

8.           Boissier, E., et al., Haemolysis index: validation for haemolysis detection during extracorporeal membrane oxygenation. Br J Anaesth, 2020. 125(2): p. e218-e220.

9.           Petrova, D.T., et al., Can the Roche hemolysis index be used for automated determination of cell-free hemoglobin? A comparison to photometric assays. Clin Biochem, 2013. 46(13-14): p. 1298-301.

10.         Altman, D.G. and J.M. Bland, Diagnostic tests 2: Predictive values. BMJ, 1994. 309(6947): p. 102.

11.         Extracorporeal Life Support Organization, ELSO Guidelines for Cardiopulmonary Extracorporeal Life Support. 2017: Ann Arbor, MI, USA.

12.         Bosma, M., et al., Automated and cost-efficient early detection of hemolysis in patients with extracorporeal life support: Use of the hemolysis-index of routine clinical chemistry platforms. J Crit Care, 2019. 51: p. 29-33.

Author Response

  1. Pearson correlation is not the correct methodology to assess the agreement between two analytical methods. Even if the Pearson correlation coefficient is an easy and fast way to establish a mathematical relation between two methods, it is widely admitted that it is not a reliable tool to assess the equivalence (the agreement) between two methods [1-3]. A perfect correlation could exist between two methods while they are totally not interchangeable. I agree with the authors that HI and PFH correlate, but this does not indicate that HI could replace PFH assay. Correlation is not agreement! Others statistical methods like those described by Bland & Altman [4], or Lin [5], have to be used. In addition, diagnostic tests have to be characterized in terms of internal validity (sensitivity, specificity, positive predictive value, negative predictive value) and this part is totally lacking in the manuscript.

Thank you for your helpful comment and your suggestion to add a Bland Altman analysis. We completely agree that a good correlation between two tests does not necessarily mean that tests can be used interchangeably. Two Bland Altman analyses are added to revised version of the manuscript, one for samples with low HI, and one for samples with higher HI, using a cutoff of 20. We also performed ROC analyses, first for the whole cohort, and then for patients under ECMO or mechanical circulatory support, and calculated sensitivity, specificity, positive predictive value, negative predictive value of HI to predict PFH > 10 mg/dl.

The Bland Altman analysis shows bias of 0.3 ± 0.2 and an upper and lower limit of agreement of 5.2 ± 0.2 and 4.6 ± 0.3 for samples with HI ≤ 20––bias and limits were –16.0 ± 3.4, and 13.8 ± 5.9 and –45.9 ± 5.9 for samples with HI > 20. The ROC analyses demonstrated strong performance, with the curve indicating excellent discrimination, in the whole cohort as well as in patients under ECMO or mechanical circulatory support.

We added the following to the text of the manuscript:

‘ROC analyses demonstrated strong performance of HI, with the curve indicating excellent dis-crimination, in the whole cohort (area under the ROC of 0.969) as well as in patients under ECMO or mechanical circulatory support (area under the ROC of 0.988).’ (abstract) (Page 1; line 33)

‘We performed Bland Altman analyses for HI measurements ≤ 20 and > 20, based on previous recommendations (19), and constructed receiver operating characteristic (ROC) curves to calculate sensitivity, specificity, and the positive and negative predictive value (PPV and NPV) of HI to detect hemolysis defined by PFH > 10mg/dl […]). (Page 3; line 126)

‘Bland Altman analysis plots are shown in Figure 3. For samples with HI ≤ 20, bias was 0.3 ± 0.2, the upper and a lower limit of agreement were 5.2 ± 0.2 and 4.6 ± 0.3. For samples with HI > 20, bias was –16.04 ± 3.4, and the upper and lower limit of agreement were 13.8 ± 5.9 and –45.9 ± 5.9.’ (results) (Page 6; line 123)

‘ROC curves are shown in Figure 4. The ROC analyses demonstrated strong performance, with the curve indicating excellent discrimination, in the whole cohort as well as in patients under ECMO or mechanical circulatory support. Test characteristics are presented in Table 3. Using a cutoff for HI of 8.5 results in a sensitivity and a specificity of 100% and 88.8%, and NPV and PPV of 100% and 37.8%, in patients under ECMO or mechanical circulatory support. Increasing the cutoff to 9.5 improved the specificity and PPV somewhat, but at the price of a lower sensitivity and lower NPV in these patients.’ (results) (Page 6; line 208)

‘The Bland Altman analysis provided a quantitative estimate of how closely the values of HI and PFH lie. In samples with HI ≤ 20, the findings suggest that HI and PFH can be used interchangeably.’ (discussion) (Page 7; line 259)

‘Sensitivity and NPV were excellent, in the entire cohort, as well as in the group with the highest risk of hemolysis. Specificity and PPV, however, were low, even after increasing the cutoff.’ (discussion) (Page 8; line 261)

The following Figures and Table were added:

‘Figure 3. Bland Altman plots for samples with HI ≤ 20 (panel A), and for samples with HI > 20 (panel B).’

‘Figure 4. ROC curves for the whole cohort (panel A), and for patients under ECMO or mechanical circulatory support (panel B).’

‘Table 3. Test Characteristics of HI to Predict PFH > 10 Using Two Different Cutoffs’

Of note, Table 3 in the original submission has been relabeled, and is now Table 4.

  1. HI original purpose is not to precisely determine the concentration of plasma-free hemoglobin but rather to detect (with a good sensitivity) a level of hemolysis that is susceptible to negatively interfere with analysis performed downstream. The degree of hemolysis susceptible to interfere with downstream analysis in the lab is not the same that the degree of hemolysis susceptible to cause damages on patients. This is why ‘HI’ is not called ‘assay for plasma free hemoglobin.’ This is also why ‘HI’ has no In Vitro Diagnostic (IVD)-FDA or Conformity for Europe (CE) mark approval for determination of plasma-free hemoglobin.

Thank you for your comment. We fully agree that the purpose of HI is to detect hemolysis which may interfere with certain analysis, i.e., meant for use in the laboratory to detect preanalytical impairments of samples. We also agree that the degree of hemolysis susceptible to interfere with downstream analysis in the lab is not the same as the degree of hemolysis susceptible to cause damages on patients. However, this laboratory parameter could be useful to alert clinicians that hemolysis is possibly occurring, which should trigger additional actions.

We discuss this in the revised version of the manuscript as follows:

‘The original purpose of HI is to detect hemolysis which may interfere with certain analysis, i.e., it is meant for use in the laboratory to detect preanalytical impairments of samples. It is important to note that the degree of hemolysis susceptible to interfere with downstream laboratory analysis is not the same as the degree of hemolysis susceptible to cause damages on patients. However, this laboratory parameter could be useful to alert clinicians that hemolysis is possibly occurring, which should trigger additional actions.’ (discussion) (Page 8; line 276)

  1. HI, regarding its potential to quantify hemolysis in the ECMO and circulatory support settings has been associated with multiples drawbacks, preventing a simple and direct applicability to the clinical context. Sadly, your paper does not cite any reference dealing with such limits, leading to potentially biased overview and analysis of the question.

We apologize for this omission. We added the drawbacks in the revised version of the manuscript. In the limitation–section we specifically address the limitations you mentioned, as follows:

As example (see [6]): HI are not comparable within manufacturers as remarkably demonstrated by Calvaresi et al. on Abbott, Roche, and Beckman platforms [7];

‘In our study, we employed a single blood chemistry analyzer, the Roche Cobas C 702. It should be noted that HI measurements are not standardized across various laboratory platforms, potentially contributing to discrepancies in results across studies. While one study suggests that the variance in HI measurements could be acceptable […], another study suggests the contrary […].’ (discussion) (Page 9; line 326)

HI reliability is impaired in case of icteric and lipemic samples, both situations that are not exceptional in patients hospitalized in critical care unit with ECMO support (liver dysfunction, propofol infusion, parenteral nutrition)

‘The reliability of HI measurements is impaired in cases of icteric and lipemic samples, conditions that are automatically displayed by the blood chemistry analyzer. In our analysis, we excluded patients with icterus, and none of the samples was lipemic.’ (discussion) (Page 9; line 333)

Some authors reported that HI measurement demonstrated poor recovery when applied to commercial hemoglobin quality control material, questioning the ability of the method to be calibrated [9];

‘In addition, quality control of HI and calibration of analyzers can be limited by problems with commercially available quality control samples, another challenge that deserves a solution from the suppliers […]’ (discussion) (Page 9; line 330)

HI may have a very poor negative predictive value (NPV), with a cutoff set to 500 mg/L, Calvaresi et al. observed NPV of 57.6% for one analyzer and 80.9% for another one [7]. Even if one could argue that this low NPV is here in relation with a high prevalence of the targeted anomaly (hemolysis), such performances are not suitable for a clinical use.

We agree; however, in our study, NPV was actually good, and we had more concerns regarding PPV. See also our response to your comment 1, above.

When considering low level of hemolysis (HI < 20), data presented here clearly indicate a low agreement with PFH assay. For some patients (panel A), a HI value equal or lower to 15 could be associated with a PFH level close to zero or higher than 200 mg/L, definitely confirming that a robust and dedicated assay is better than a non-specific method, originally devoted to another purpose (i.e. to detect with a good sensitivity a level of hemolysis that is susceptible to interfere with analysis performed downstream).

We added Bland Altman analyses as suggested by you; see also our response to your comment 1, above.

  1. The design of the study is unusual to allow a robust analysis. Why did you include patients that are not at risk of mechanical hemolysis (no extra-corporeal or mechanical circulatory support)? This population represents here in your cohort a vast majority (62%) of the patients. In this population, by definition, hemolysis events are present at very low frequency. You then enter this population within the whole analysis, to assess the correlation of HI, PFH and LDH. By this way, you introduce a bias where you analyze couples of measures obtained from 2 different populations: one (minority) with a high prevalence of hemolysis and one (majority) with a low prevalence of hemolysis. As negative and positive predictive values depend on the prevalence of the disease in the studied population[10], such a design is ambiguous.

Thank you for your comment and pointing out the possible bias. Following your advice, we constructed ROC curves, for the entire cohort, and for patients under ECMO or mechanical circulatory support. See our reply to your comment 1 above, and the results of the additional analyses in the extra Figures and Table.

  1. In your paper, the performance of the HI to correctly correlate with PFH is lower when < 20 (+/- 200 mg/L in Roche automate), meaning that some patients (as previously exposed) could have no hemolysis or a significant hemolysis (>200 mg/L of pFH). ELSO guidelines [11] recommend to maintain PFH below 100 mg/L and Bosma et al. [12] observed that a repeated HI >20 was a predictor of mortality. These 2 points clearly demonstrate that a HI cut-off of 20 is too high (e. to belated) and that a more sensitive way of assessing hemolysis is required to prevent / treat (earlier) the causes of this hemolysis.

Excellent point. The constructed ROC curves suggested lower cutoffs, and their performance is shown in the new Table 3.

Reviewer 2 Report

Comments and Suggestions for Authors

I sincerely thank the editor for the opportunity to review this paper. The authors assessed the correlation between the Hemolysis Index (HI) and PFH (probably the gold standard) and LDH in ECMO patients. HI correlated well with PFH and may represent a readily available and cost-effective test to monitor hemolysis. I believe this is a very important and practice-changing study; the authors should be commended. The manuscript is clear and well-written.

I have a few comments:

  1. I would add some detail in the introduction section about how HI is measured in the laboratory. For example, you could include information from Dolci 2013 (Chem Acta): 'HI is a calculation based on absorbance measurements performed on serum/plasma at different wavelengths, providing a semi-quantitative estimate of hemolysis detected in the sample.'
  2. Regarding Figure 2, Panel D: I believe the R value is incorrect. Additionally, I suggest adjusting the Y-axis scale, perhaps setting the upper limit at 50?
  3. In the discussion section, consider adding that LDH may increase for various medical reasons (e.g., liver disease, anemia, heart attack, bone fractures, muscle trauma, cancers, infections, etc.). Based on your findings, PFH may be disregarded in the future, but LDH monitoring may have other clinical implications.
  4. A major limitation could be the comparability of HI values between different ECMO centers with different laboratories. This should be added among the limitations."

These suggestions aim to enhance clarity and precision in your communication.

Author Response

  1. Line 112. ‘using Pearson’ should be replaced by ‘using Pearson method’

Thank you for your comment. We changed the sentence to:

‘Correlations between HI, PFH and plasma LDH were calculated using Pearson correlations coefficients.’ (methods) (Page 3; line 120)

  1. Line 119-124: I don’t understand the extrapolation that is performed. Why do you screen patients for hemolysis, once a day, when they are not receiving mechanical circulatory support? This is quite odd. In addition, why did you perform a monitoring of hemolysis every 6 hours (4 times per day) on patients at risk of hemolysis? Such a sampling frequency is quite high and unusual as a routine practice in our country and others. Could you justify such a habit? The cost reduction mentioned in your work is clearly impressive but it could be argued that you clearly did not apply the most economical strategy in routine.

Thank you for your question. In our ICU, we deem monitoring hemolysis as an essential strategy for patients undergoing ECMO or mechanical circulatory support. We started using HI to evaluate the hemolysis status in these patients, as this measure came at no additional costs. Due to this convenience, this policy has been extended to patients weaned off ECMO or mechanical circulatory support, albeit with a lower frequency of measurements. We believe this practice is not unique to our center; however, it is evident that the cost-effectiveness may vary between centers when employing different sampling frequencies.

We added this explanation to Discussion, as follows:

In our ICU, we deem monitoring hemolysis as an essential strategy for patients undergoing ECMO or mechanical circulatory support. We started using HI to evaluate the hemolysis status in these patients, as this measure came at no additional costs. Due to this convenience, this policy has been extended to patients weaned off ECMO or mechanical circulatory support, albeit with a lower frequency of measurements. We believe this practice is not unique to our center; however, it is evident that the cost-effectiveness may vary between centers when employing different sampling frequencies.’ (discussion) (Page 9; line 315)

  1. I would add some detail in the introduction section about how HI is measured in the laboratory. For example, you could include information from Dolci 2013 (Chem Acta): 'HI is a calculation based on absorbance measurements performed on serum/plasma at different wavelengths, providing a semi-quantitative estimate of hemolysis detected in the sample.'

Thank you for your valuable comment, we added these words to the Introduction:

‘HI is a calculation based on absorbance measurements performed on serum/plasma at different wavelengths, providing a semi-quantitative estimate of hemolysis detected in the sample […].’ (introduction) (Page 2; line 65)

  1. Regarding Figure 2, Panel D: I believe the R value is incorrect. Additionally, I suggest adjusting the Y-axis scale, perhaps setting the upper limit at 50?

Thank you for these suggestions; corrected

  1. In the discussion section, consider adding that LDH may increase for various medical reasons (e.g., liver disease, anemia, heart attack, bone fractures, muscle trauma, cancers, infections, etc.). Based on your findings, PFH may be disregarded in the future, but LDH monitoring may have other clinical implications.

Thank you for your comment. We modified the mentioned paragraph in our revised version of the manuscript as follows:

‘Based on our findings, PFH may be disregarded in the future, but LDH monitoring may have other clinical implications. Indeed, LDH may increase for various medical reasons, including liver disease, anemia, cardiac injury, muscle trauma, cancers, and infections. (discussion) (Page 9 line 322)

  1. A major limitation could be the comparability of HI values between different ECMO centers with different laboratories. This should be added among the limitations.

Thank you for pointing out this limitation. See our reply to one comment by reviewer 1. In response to that comment we added the following words to the revised version of our manuscript:

‘In our study, we employed a single blood chemistry analyzer, the Roche Cobas C 702. It should be noted that HI measurements are not standardized across various laboratory platforms, potentially contributing to discrepancies in results across studies. While one study suggests that the variance in HI measurements could be acceptable […], another study suggests the contrary […].’ (discussion) (Page 9; line 326)

Round 2

Reviewer 1 Report

Comments and Suggestions for Authors

I thank the authors for their work, which has now clearly improved the manuscript. I appreciate the effort made by the authors to increase the level of detail and the re-investigation of their data.

Nonetheless, I remain embarrassed by the global tone of the message that is provided by the authors. As correctly now depicted and described by the authors themselves, Hemolysis Index is a very poor diagnostic test! The value and usefulness of a diagnostic test for clinicians do not only rely on intrinsic parameters like specificity and sensitivity. That is rather the aim of the PPV (probability of having a hemolysis if the test is positive) and NPV (probability of having no hemolysis if the test is negative) of a test, the more relevant parameters for clinicians and clinical decisions.

Regarding your results, the PPV is very low! Actually, no diagnostic test with such a value is on the market! With a cutoff of 9.5 and a patient under ECMO, a PPV of 54% means that the probability of having a hemolysis (PFH >10) if HI is higher than 9.5 is 54%. With a cutoff of 8.5, it falls to 34%. Is it best to flip a coin? Clinical decisions can not be made on this basis.

Authors have therefore to temperate their message and insist on the fact that HI could be a good way to rule out hemolysis (very good NPV) but in no way could replace PFH in a given range of value because PPV is not high enough. Line 322, it seems therefore difficult to write “based on our findings, PFH may be disregarded in the future”. Your conclusions must be also in adequation with your results (temperate).

Author Response

Thank you for this comment – we actually think we are on the same line here, as we do not intend to suggest utilizing HI as a diagnostic test, but only as a monitoring tool. Indeed, HI proves inadequate as a diagnostic tool, as clinicians cannot depend on a test with a low PPV, as evidenced by our analysis. Thus, clinical decisions should not be grounded solely on HI measurements. However, its high NPV indicates a strong likelihood of absence of hemolysis when the test is negative, enabling the exclusion of hemolysis. Thus, we assert that a ‘monitoring strategy’ employing HI may be preferable over those relying on PFH or plasma LDH for detecting hemolysis.

In essence, we emphasize its utility in monitoring for hemolysis rather than diagnosing it.

We made several changes in the second revised version to make this statement clear We made several changes in the second revised version to make this
statement clear, including the following:

‘Although the negative predictive value of HI for predicting PFH levels >
10 mg/dL was high, its positive predictive value was found to be poor at
various cutoffs. […] a strategy utilizing HI may be preferable for
monitoring for hemolysis […]. The PPV of HI, however, is unacceptably
low to use it as a diagnostic test.’ (abstract) (Page 1, line 36)
‘Using a cutoff for HI of 8.5 results in a sensitivity and a specificity of
100% and 88.8%, an excellent NPV but a poor PPV of 100% and 37.8%,
respectively in patients with ECMO or mechanical circulatory support.’
(results) (page 6, line 217)
‘It is crucial to emphasize that while HI may serve as a monitoring tool
for hemolysis, our findings unequivocally demonstrate its insufficiency
as a diagnostic instrument. The inadequacy of HI as a reliable diagnostic
tool is evident from our analysis, rendering it unreliable with a notably
low PPV. Consequently, clinicians must exercise caution and refrain from solely relying on HI measurements in making critical clinical
decisions.’ (results) (page 9, line 332)
In Conclusion: ‘HI may be an attractive alternative for PFH or plasma
LDH in respect of monitoring for hemolysis in the ICU. The PPV of HI,
however, is unacceptably low to use it as a diagnostic test.’ (results)
(page 9, line 358)
